# Impact and Assessment of Suspension Stiffness on Vibration Propagation into Vehicle

**DOI:** 10.3390/s23041930

**Published:** 2023-02-09

**Authors:** Rafał Burdzik

**Affiliations:** Department of Road Transport, Faculty of Transport and Aviation Engineering, Silesian University of Technology, 40-019 Katowice, Poland; rafal.burdzik@polsl.pl; Tel.: +48-32-603-41-16

**Keywords:** vibration propagation, coil spring, suspension stiffness, vehicle

## Abstract

The impact of transport-induced vibrations on people is a particularly important problem. Sudden or intensifying vibration phenomena of a local nature may compromise safety, especially in transport. The paper addresses the results of research on the impact of spring stiffness parameters on the propagation of vibrations in the vehicle structure using simple amplitude and frequency measures. The use of the developed method of selective multi-criteria analysis of frequency bands made it possible to compare the vibrations recorded in the vehicle with a new or worn coil spring. The results of the present study allow the development of a large data base in which all signals are classified by the exploitation parameters and location of the propagation of vibration in the vehicle. The most important findings and achievements of the presented study are the testing of actual suspension components with real damage under controlled conditions, the identification of the vibration propagation path from the wheel to the driver and passenger feet, the quantitative comparison of vibrations affecting humans in the vehicle (through the feet), and the frequency decomposition of vibration for selected bands. These findings improve the proper interpretation of the developed measures and, as a result, the difficulties in using this knowledge at the engineering level, for example, in the design and construction improvement stage. Therefore, innovation points and engineering significances are a method of selective multi-criteria analysis of frequency bands and have potential applications in diagnostics and the design of suspension systems and in terms of passengers’ comfort.

## 1. Introduction

Sensing technology and data interpretation in machine diagnosis and system condition monitoring are foundations of information collection on the current technical condition of a technical system. Most of the time these are topics of focus in a wide range of technical diagnostics [1,2,3,4] and medicine [5,6,7]. Therefore, system condition monitoring is a fundamental process for decision making protocols in all mechanical or biological systems. The control and steering of all systems determine their operational and functional activities. The areas of technical diagnostics in theoretical approaches and industrial applications have been presented in [1]. An interesting trend in sensing technologies is MetaSurfaces. The current development of artificial 2D materials with peculiar electromagnetic properties has opened new possibilities in technical diagnostics [2]. MetaSurfaces are artificial materials with exotic electromagnetic characteristics composed of dielectric inclusions arranged in a grid configuration much smaller than the operative wavelength [8,9]. Metasurfaces take up less physical space than complete three-dimensional metamaterial structures engineered by arranging a set of small scatterers in a regular array throughout a region of space [9]. These advantages make metasurfaces useful in both technical and medical diagnostics [5]. Biosensors are also of great use in medical diagnostics. Biosensor technologies are mainly based on the analysis of whole blood samples from patients. The main task is to find the relationship between symptoms and disease with disease-specific biomarkers. A paper [6] reported recent advances in the field of 3D biosensors for clinical applications. Three-dimensional (3D) biosensors are the results of the development of advanced biomaterial methods, biochemical tools, and micro/nanotechnology approaches. Other examples of sensing technology are non-invasive external brain monitoring devices that are used as sensors for stroke and other neurological conditions. A review and summary on non-invasive brain monitoring devices have been presented in [7]. It states that some of the potential applications of these devices include distinguishing stroke from non-stroke, which could transform the current diagnostic and treatment paradigm for this disease.

As a rule, the application of diagnostics, regardless of whether it is in the field of medicine or technology, consists of selecting the appropriate method, using appropriate measuring devices, and using effective signal processing to extract information and assess the condition. Continuous or periodic monitoring of the condition enables the detection of inconsistencies (defects) in the early stages of development, preventing the occurrence of emergencies, defects, or diseases in the case of medicine [10,11]. It is also possible to build large knowledge bases (Big Data) to define long-term forecasts that improve the management of technical systems operation. A paper [12] presented a study using a deep learning network to forecast wind turbine power based on a long- and short-term memory network algorithm. Big data is also used in intelligent transportation systems (ITS). The framework for performing big data analytics in ITS is discussed in [13]. The authors described data sources and collection methods, data analytics methods and platforms, and big data analytics application. 

In this paper, it was decided to present a different approach that extended the scope of the diagnosis. This extension consists of monitoring phenomena (signals) that affect humans and are determined by the technical conditions of the technical system. This approach extends the application of diagnostics to preventive measures, the aim of which is to eliminate or reduce the harmful effects of the technical system on humans at the source, that is, in the identified area and place of exposure [14]. Vibrations are an important factor in hazards and the impact of machines on humans. In this case, the place of impact on the human body is important, as is the place of contact between the machine and the human body. In this aspect, the areas of application of the research presented in this paper concern technical diagnostics, safety, comfort, and ergonomics. In this aspect, machine diagnosis and system condition monitoring decisions should also study the damage or decreased operating parameters on the comfort and health of operators, in this case, the driver and passengers of the motor vehicle. Therefore, an attempt was made to identify and assess the influence of the technical conditions of the suspension components on the propagation of vibrations in the vehicle. The source was assumed to be the propagation path starting from the car wheel into the floor panel in the places of transfer (transmission) of general vibrations to humans via the feet.

Vehicle vibrations should be considered as unwanted negative phenomena resulting from the contact of rolling wheels with road unevenness, which are perceived as dynamic forces transferred by vertical wheel movements to the vehicle body and, as a result, to the driver and passengers [15]. One of the key components of the vehicle in this aspect is the suspension, equipped with damping elements (shock absorbers) and elastic elements (springs). The proper functioning of the suspension elements should guarantee the correct driving of the vehicle and combining the unsprung and sprung masses [16,17]. In addition, it is responsible for ensuring constant contact of the wheel with the road surface, as well as comfort, understood as minimizing the vibrations transmitted to the people in the vehicle [18]. The path of propagation from the wheels, through the supporting elements of the suspension and vehicle frame, to the floor and equipment, including seats, reaches the driver and passengers, causing exposure to vibrations. The frequencies and the dose of vibrations depend on the non-linear characteristics of the elements on the propagation path [18]. The constant increase in traffic in cities and urban areas causes an increase in human exposure to noise and vibration [19]. Most of research and publications in this area are concerned with the impact of vibration and noise on the external environment [20,21]. Much less research has been shown on the impact of vibrations on people in cars, buses, or other means of urban transport [22]. The impact exerted by transport-induced vibrations on people is a particularly important problem. These vibrations directly affect the comfort and indirectly the safety of driving or riding in a vehicle. Prolonged exposure to vibrations at specific frequencies can affect drivers’ perception and vision. 

Vibrations in a means of transport not only have a negative impact on drivers and passengers [23], but they are also a source of evolutionary degradation of roads, bridges, and other elements of transport infrastructure [24]. Vibrations transmitted to humans through the hands (local vibrations) or feet and torso (general vibrations) to which a person is exposed for a long time adversely affect human health, especially when the vibration frequencies coincide with the natural vibration frequencies of the body organs [25,26]. As the main factors determining the pathological effects of exposure to vibrations, the frequency of vibrations, exposure time, and body part of vibration penetration should be mentioned [27]. The negative influence of vibration acceleration on human are called kinetosis [28]. Papers [27,28] have confirmed that long-term exposure to vibrations can even indirectly affect diseases of the cardiovascular, respiratory, digestive, and musculoskeletal systems. A large professional group exposed to vibrations is drivers and operators of means of transport (e.g., construction machinery or even overhead cranes). General vibrations are transmitted through the seats and floor of the means of transport, and local vibrations are transmitted mainly through the steering wheel or gear lever [29,30]. 

However, it should be pointed out that the principal tasks of vehicle suspension are to ensure safety and control of the vehicle’s driving (steerability) by maintaining contact between the wheels and the road surface. The correct technical condition of the vehicle is a fundamental factor in road safety because the technical failure of vehicles can cause road accidents [31]. Regulations and systems were developed to inspect the technical conditions of vehicles operating according to traffic safety parameters [32]. Therefore, in previous research and publications, the authors were focused on developing a diagnostic method and advanced signal processing algorithm and classification system for the identification of detailed and combined shock absorber defects [33]. However, this article focuses on issues related to the measurement and analysis of the impact of the technical condition of a selected suspension element on vibrations that are transferred to the driver and passengers through the foot contact with the floor of the car cabin.

The multi-channel recording of synchronous vibration signals along a specific path of vibration propagation from the excitation on the wheel, through the elements of the suspension system to the vehicle floor, can be considered as a new approach to the research issues presented in this paper. This approach makes it possible to track changes in the structure of vibration signals by wave propagation to the place where vibrations affect the feet of passengers. In order to present the engineering application potential of this method, the results of experiments were presented, during which vibrations of a system with various elastic elements (coil springs) were recorded. Thanks to the use of signal estimators and analysis of frequency components, it was possible to assess the stiffness of the spring element of the suspension and to analyse its impact on vibrations affecting the passengers.

## 2. Research Methods

When vibrations are examined, the main purpose of the research should be indicated. Vibrations are often used for diagnostic purposes as information carriers. In the case of this article, the main purpose of the research is to assess the impact of the elastic parameters of the vehicle suspension on vibrations transferred to the vehicle floor, at places of contact with the feet of passengers and the driver. In this aspect, two types of vibrations should be distinguished: general and local vibrations. The category of vibrations in this case depends on the place where the vibrations enter the human body. This division in the case of vibrations occurring while driving a car is shown in Figure 1.

The main determinants of stiffness vibration are mass, stiffness, and damping. Therefore, properties of elementary suspension components are very important. As a consequence, driving comfort and safety depend on the ratio of sprung and unsprung masses, the damping force of shock absorbers, and the stiffness of the coil spring [34,35], as well as the parameters of car seats and even the type and elasticity characteristics of tires [36]. Exemplary studies of the impact of tires on whole-body vibrations are presented in [37]. The methods and measures to assess the impact of vibrations on comfort have been formalised in the ISO 2631-1 standard [38]. The development of reliable vibration comfort in vehicles is still a scientific challenge due to the subjectivity of vibration perception. Some papers [39,40] have presented several proposals for measures of vibration comfort for child car seats. Based on the analysis of the literature, the author concluded that there is a significant number of papers on vibration comfort in means of transport related to seat vibrations. Therefore, as part of the investigation, it was decided to record floor vibrations at the place where the vibrations penetrate through the feet.

The scope of the research included further active experiments consisting of introducing suspension elements with recognized parameters, e.g., spring characteristics of a coil spring and conducting vibration tests. The tested passenger car belongs to a group of very popular cars in Europe. To excite the entire system (car) to vibration, an original test stand with kinematic excitation of vibrations was used. To control the vibrations, a frequency converter was used, on which the excitation band was defined, which contains the most important resonant frequencies of the unsprung and sprung masses.

During the research, vibration accelerations were recorded at the chosen points of the car body and suspension elements. The first phase of vibration transfer was defined as input force (acceleration of the excitation plate) and vertical vibrations of unsprung elements (suspension arm). This represents the damping properties of tires. The next location of the measurement chain was the vehicle body (upper shock absorber mounting) and was recorded coaxially. This represents the properties of the suspension system (spring and shock absorber), that are strongly dependent on the stiffness and damping of the complex suspension system. In order to analyse the distribution of vibrations as the sources of general vibration exposure of the driver and passengers, vibrations of the floor panel were recorded at locations where passengers rested their feet on the floor (four locations in the measurement chain). These locations represent general vibrations that penetrate the humans inside the vehicle. The location of all measurement points is shown in Figure 2. Components of investigated front suspension system with replacing coil spring have been depicted in Figure 3.

The parameters of measurement system used in research have been depicted in Table 1. 

In order to force vehicle vibrations by introducing the wheel to oscillating vertical movements, a test stand equipped with a kinematic vibration exciter was used. The car drove the wheel onto the ramp (plate) of the stand, which then activated the exciter, and the excitation plate stimulated the wheel to vertical movements according to a specific excitation force profile. The exciter controlled the inverter with a programmed excitation frequency programme to a certain frequency. The test plan forced the vehicle to vibrate above the resonant frequency of the unsprung masses, passing through the bands characteristic for the resonance of sprung masses (1–3 Hz) and unsprung masses (8–15 Hz). Thanks to this, it was possible to analyse vibrations in terms of safety and comfort. The frequency and amplitude functions of the input force (exciter plate) are shown in Figure 4.

The properties of coil springs change due to the operating period. It was assumed that the main factor in vibration transfer related to the springs is time and condition of operation of the vehicle suspension. The experiments were carried out on a passenger car with a new and used (worn-out) suspension spring. The force vs. deflection characteristics of researching coil springs are presented in Figure 5. The influence of stiffness on the dynamic response of a mechanical system is also dependent on the initial static load of this system. In this case, the differences in the vibrations transmitted by such a system can be even more pronounced with a change of the spring and additional (extra) load. Therefore, during the research experiments, vibration recordings were carried out with an additional (extra) load on the test vehicle. 

Each measurement series consisted of several measurements for the same excitation parameters and spring stiffness characteristics, during which vibration acceleration signals were recorded synchronously at all measurement points.

## 3. Results and Discussion

The assumption of the analysis of the results was to assess the impact of the stiffness parameters on the propagation of vibrations in the vehicle structure using simple amplitude and frequency measures. This is due to the large number of articles which used advanced and often very time-consuming mathematical algorithms and signal processing methods. A complex methodology for the multidimensional analysis of nonstationary vibration signals has been presented in [42]. An even more advanced algorithm for computing the averaged infogram for local damage detection in the presence of non-Gaussian impulsive noise was studied in [43]. Another approach for advanced signal processing in machine diagnostics as the method for long-term data segmentation in the context of machine health prognosis is presented in [44]. Signal processing methods can be used for any kind of signals, for example, the parameters of the engine operating condition and course of the combustion process, as presented in [45].

Often, the high accuracy of these methods and the advanced separation of signal components cause difficulties in the unambiguous interpretation of the developed measure and, as a result, difficulties in using this knowledge at the engineering level, for example, at the design and construction improvement stage.

The waveforms of vibration recorded in the course of testing have been illustrated in Figure 6. It presents comparison on vibration signals recorded when a new sprig was placed in suspension (blue colour) and used (worn-out) spring (orange colour). The consecutive channels have been assigned the following numbers:−channel 2—accelerations of the excitation plate vibrations (input force);−channel 3—accelerations of the suspension arm vibrations; −channel 4—accelerations of the vibrations of the upper shock absorber mounting;−channel 5—accelerations of the floor panel vibrations under the driver’s feet; −channel 6—accelerations of floor panel vibrations under the front passenger feet; −channel 7—accelerations of floor panel vibrations under the rear left passenger feet; −channel 8—accelerations of the floor panel vibrations under the rear right passenger feet.

**Figure 6 sensors-23-01930-f006:**
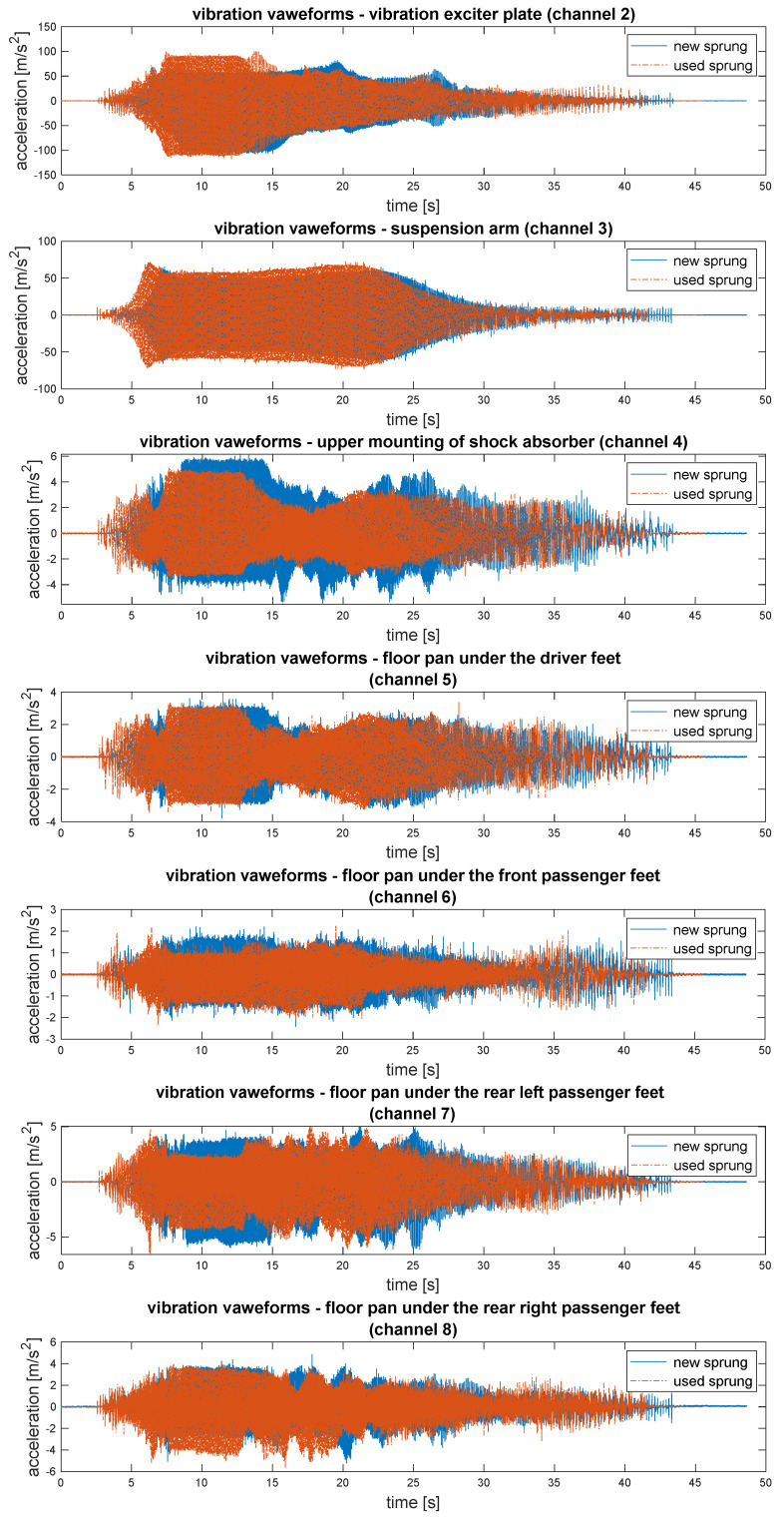
Comparison of vibration of the tested vehicle with new and used (worn-out) suspension springs showing the distribution of vibrations in the vehicle structure.

Even raw signals enabled precise observation and comparison of vibration time functions. Thus, the evaluation of exploitation parameters on the vehicle to propagate vibrations into vehicle structure is possible. For the preliminary analysis of the influence of the investigated parameters on the level of vibrations occurring in vehicle structure, the resonance windows were identified. According to mechanical vibration theory, the impact of mechanical system properties (e.g., stiffness) on the generation and propagation of vibrations is significant during the passing of resonance frequencies. The distribution of the maximum values of the vibration acceleration from suspensions with new and worn springs are depicted in Figure 7. These figures present complete results for the research with an additional (extra) load in the tested vehicle.

The analysis of vibrations in the resonance windows with maximum values can only be a preliminary observation. The distributions of the maximum values of vibrations recorded for the two different coil springs presented in Figure 7 show differences and some cognitive values, but they do not constitute valuable engineering knowledge. When assessing the impact on vibration comfort, extended analyses should be carried out, which will take into account measures of signal energy and frequency distribution. To analyse the frequency distribution of vibrations, a mathematical transformation of the time series into the frequency distribution should be performed. Fast Fourier Transform (FFT) is commonly used for this purpose.

A certain novelty of the presented research is the analysis of the frequency structure of synchronously recorded vibrations on the vehicle floor in places where the vibrations penetrate the people in the vehicle through the feet. In addition, to analyse the vibration propagation, taking into account suspension parameters, vibration spectra were determined from the source of vibration through the suspension arm (representing the unsprung mass) and the upper shock absorber mounting, as the first point of the sprung mass. The sample results are shown in Figure 8.

In order to analyse the degree of vibration isolation and dampening by the suspension system, including the coil spring, the vibration spectra recorded on the suspension arm (unsprung mass) and the upper shock absorber mount (body, unsprung mass) should be compared. This comparison is visible when magnified for the resonant band of unsprung masses, 10–14 Hz (upper graph in Figure 9). A high degree of vibration dampening in this band was visible. 

Furthermore, Figure 9 shows selected characteristic frequency bands and a comparison of the vibration signal spectra recorded on the vehicle floor (where the feet of passengers and the driver are placed). Based on these analyses, you can see which frequencies are dominant at different points in the vehicle. This is very important information which confirms the legitimacy of considering vibration isolating elements individually, depending on the installation location in the vehicle.

Specific frequency bands were selected for further analysis. The criterion for the definition of these bands included the dominant characteristic frequencies related to safety and comfort, as well as bands related to the natural frequencies of selected human organs that affect perception (e.g., eyes) and a sense of discomfort. Such a selection of frequency bands makes it possible to develop automatic algorithms for assessing vibration properties in a multi-criteria approach. The characteristic frequency bands adopted for further analysis are presented in Table 2.

Figure 10 illustrates the signal frequency bands assumed for the analysis of vehicle floor panel vibrations.

The next stage of the analysis involved developing a collation of maximum values of the signal spectrum amplitudes in the frequency bands selected for the technical conditions of the coil springs. The results of the analysis of the vibrations of the new and used (worn-out) suspension springs were collected in the database form. The cases examined represent the spring properties presented in Figure 5, such as the force vs. deflection characteristics of the coil springs. The following bar charts graphically present the chosen analysis results that were obtained (Figure 11 and Figure 12).

Based on the empirical studies, sprung masses resonant phenomena that occur at higher frequencies, even exceeding 5 Hz, have been identified, namely those which may cause considerable discomfort. In terms of unsprung masses, free vibration frequencies assumed values within a range from several to more than a dozen hertz (i.e., 8–18 Hz). While an automotive vehicle is moving, free vibrations of sprung and unsprung masses occur simultaneously and overlap.

## 4. Conclusions

Studies of the impact exerted by the technical condition of suspension components on the propagation of vibrations generated by a vertical motion of the road wheel in the vehicle structure are particularly important for the analysis and identification of the sources and propagation method in vehicles. The purpose of these studies was to identify factors that affect the propagation of vibrations caused by the dynamic impact of road irregularities on the wheels of a moving vehicle.

This paper presents the results of analyses of frequency-based vibration measures recorded on a vehicle floor panel for different properties of suspension coil springs and stiffness characteristics. The analysis of selected characteristic frequency bands showed that in the case of vibrations of the vehicle floor under the driver’s feet, the greatest differences for a new and worn spring occur for frequencies around 13 Hz and 193 Hz. The first frequency is especially important because it is close to the resonant frequency of the sprung masses, strongly affecting the safety and maintenance of constant contact of the wheel with the road. For the remaining sensor locations under the passengers’ feet, the greatest differences were noted for the frequency of about 21 Hz, which is correlated with the phase of constant extension and represents driving at a constant speed. This indicates the influence of the suspension stiffness on the vibration comfort of passengers during a long drive at a constant speed, for example on a motorway.

The use of the developed method of selective multi-criteria analysis of frequency bands made it possible to compare the vibrations recorded in the vehicle with a new or worn coil spring. For most of the selected bands, the estimators obtained when testing the vehicle with a worn spring were lower, with the exception of bands no. 3 (12–17 Hz) and no. 4 (21–22 Hz). It is worth noting that band no. 3 is similar to the resonance band of sprung masses, responsible for comfort, while band no. 4 is correlated with the phase of constant excitation, correlated with driving at a constant speed in the range of approx. 60–80 km/h.

The results of the present study allow the development of a large size data base in which all signals are classified by the exploitation parameters and location of the propagation of vibrations in the vehicle. It allows the analysis of local anomalies of the vibration signal, in which the most significant differences for the technical condition of the vehicle elements can be tested.

Having compared the values recorded on the unsprung masses with all other values (sprung masses), one may perform a preliminary assessment of the quality of vibration dampening and the insulation of the vehicle body to protect it against road-induced vibrations. For more in-depth conclusions advanced signal processing will be required.

The presentation of changes in single-value measures as a representation of general vibration signals provides preliminary conclusions and decisions about structural changes and reverification in the context of assessing the degree of improvement after introducing a change in the elements of the suspension system. These findings can be used in engineering practise. In addition, it is possible to extend the analyses by assessing the impact of these changes on passenger comfort and health in terms of exposure to vibrations depending on the technical condition of the suspension. However, this will require extended analyses using human/dummy experiments.

Recognizing the continuous and dynamic development in sensing technologies in transport and the possibilities and installation of sensors in various places, which is already at the production stage [46,47], it can be assumed that appropriate vibration sensors can be installed on the vehicle floor, in places where passengers come into contact with vehicle elements and where vibrations penetrate into people in order to monitor signal parameters, e.g., in terms of comfort, health risks, and exposure.

## Figures and Tables

**Figure 1 sensors-23-01930-f001:**
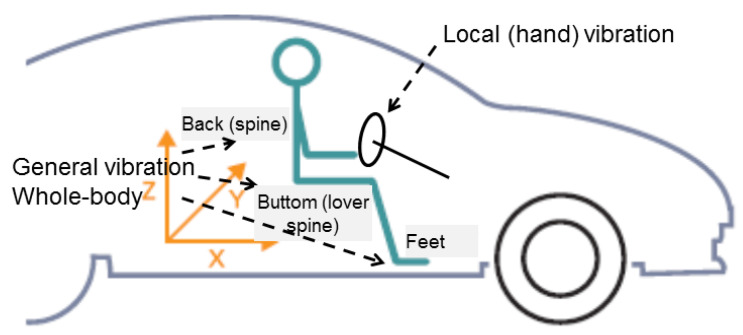
Classification of vibration in term of exposure to a human in a passenger car.

**Figure 2 sensors-23-01930-f002:**
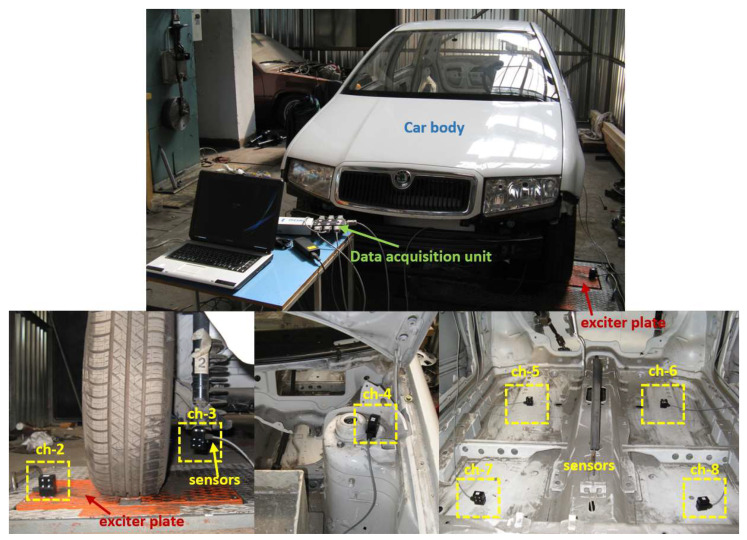
Overall experimental set-up with exciter plate and data acquisition unit and locations of vibration sensors across the structure of the vehicle tested.

**Figure 3 sensors-23-01930-f003:**
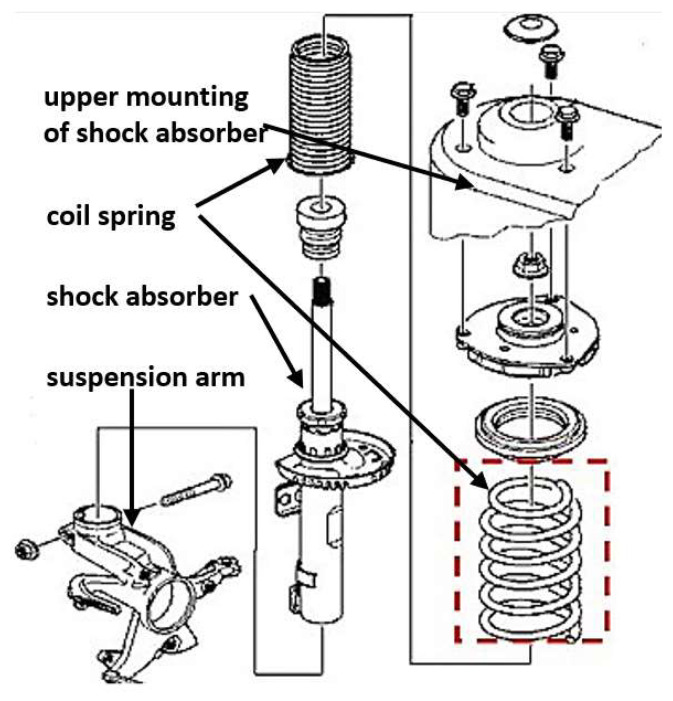
Diagram of the front suspension system with replacing coil spring (objects of investigation) [41].

**Figure 4 sensors-23-01930-f004:**
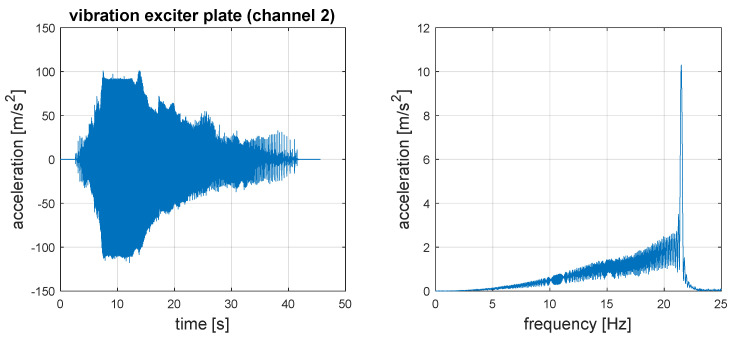
Input signal—vibration of exciter plate.

**Figure 5 sensors-23-01930-f005:**
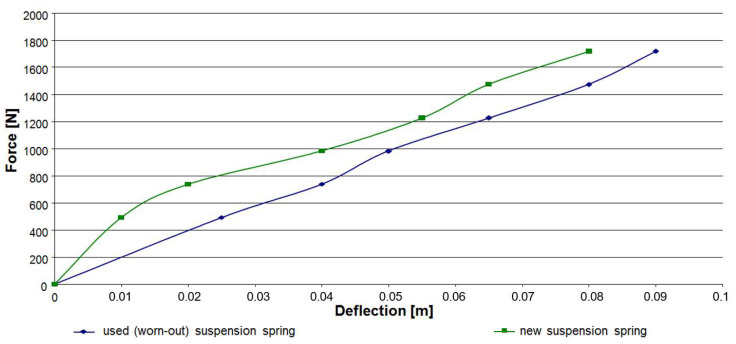
Force vs. deflection characteristics of coil springs applied during research.

**Figure 7 sensors-23-01930-f007:**
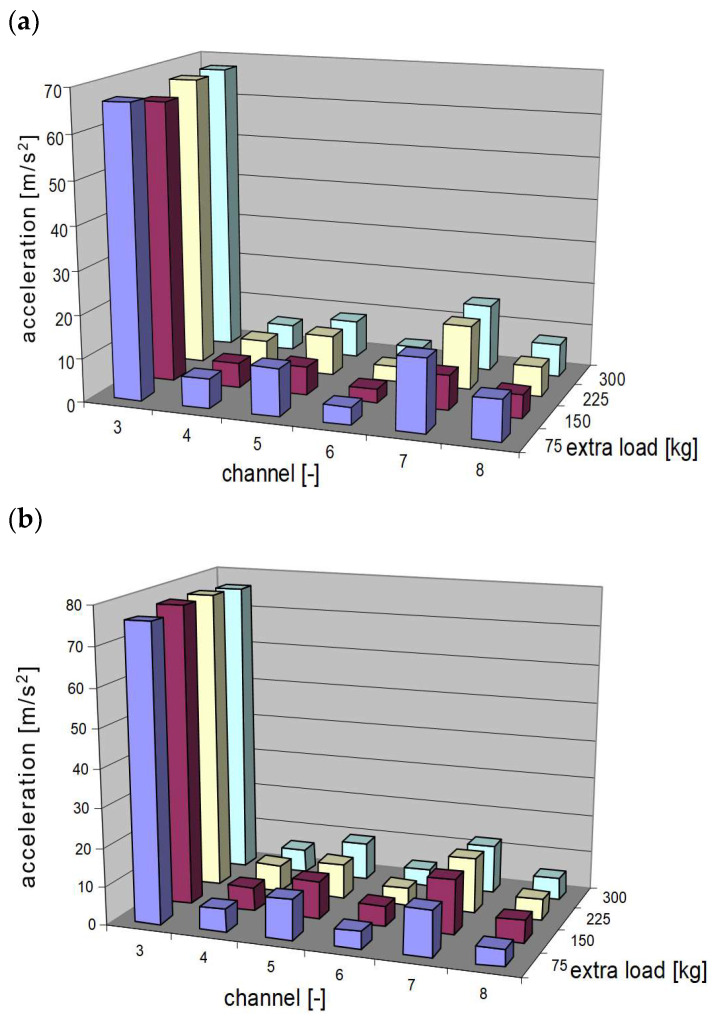
Distribution of changes in the maximum values of vibration accelerations in resonance window for a vehicle with build with (**a**) new and (**b**) used (worn-out) springs.

**Figure 8 sensors-23-01930-f008:**
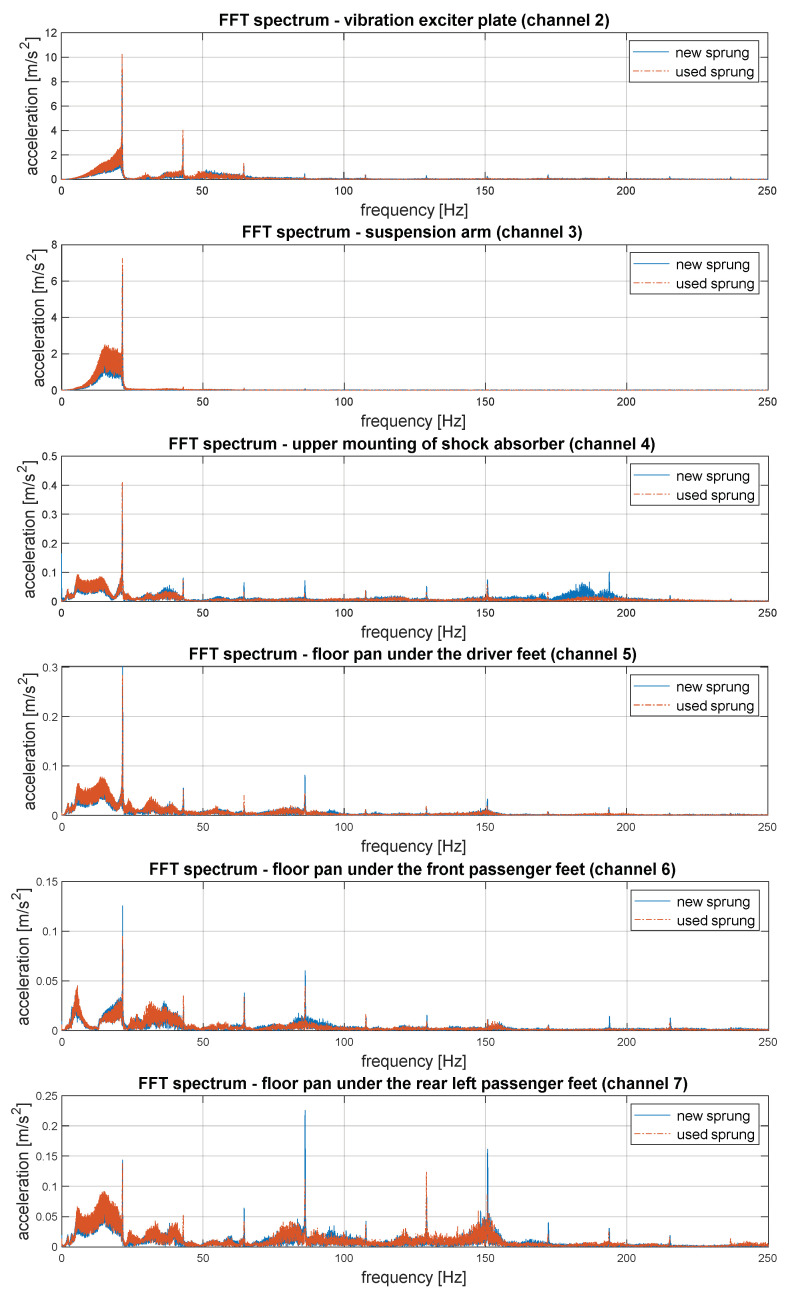
Spectrums of vibration in different location of suspension and car-body.

**Figure 9 sensors-23-01930-f009:**
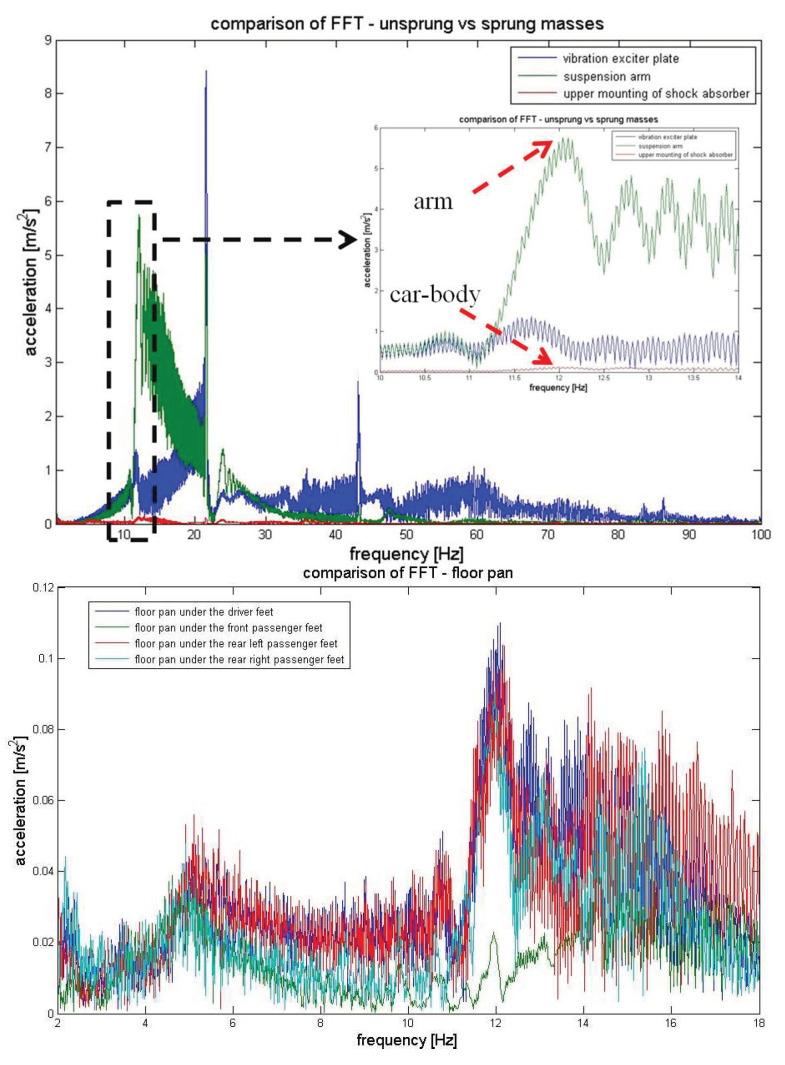
Comparison of spectrums in selected frequency bands.

**Figure 10 sensors-23-01930-f010:**
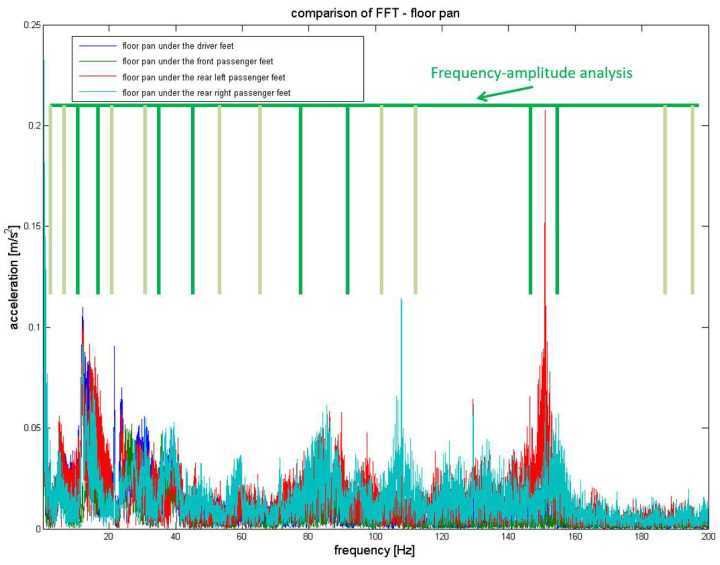
Frequency bands analysed for the recorded vehicle floor panel vibration signals.

**Figure 11 sensors-23-01930-f011:**
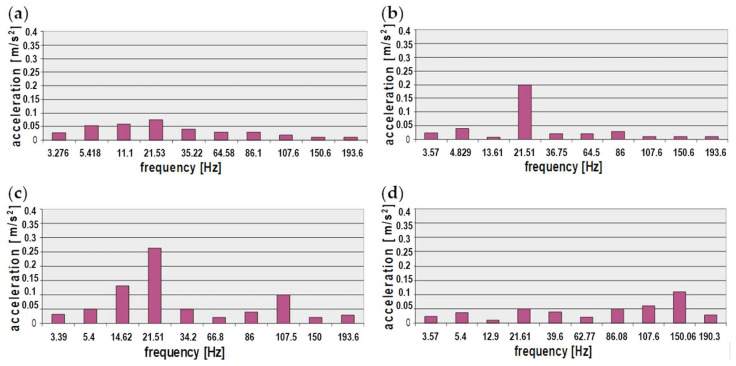
Distribution of maximum amplitudes of vibration signal spectra obtained for a floor panel of passenger car for the selected characteristic frequencies (the vehicles with build-in new suspension spring): (**a**) driver’s feet resting spot, (**b**) front passenger’s feet resting spot, (**c**) rear left passenger’s feet resting spot, (**d**) rear right passenger’s feet resting spot.

**Figure 12 sensors-23-01930-f012:**
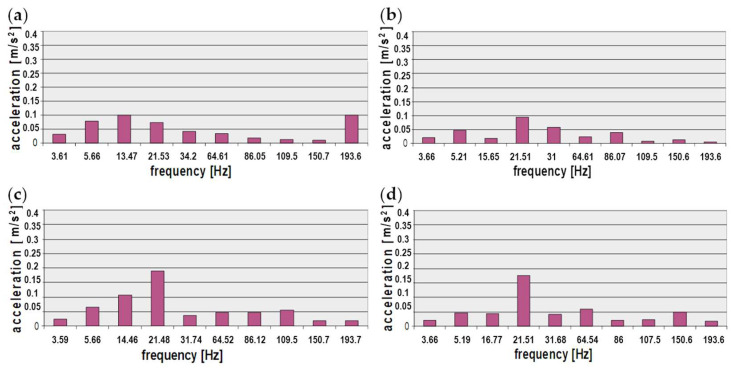
Distribution of maximum amplitudes of vibration signal spectra obtained for floor panel of passenger car for the selected characteristic frequencies (the vehicles with build-in used (worn-out) suspension spring): (**a**) driver’s feet resting spot, (**b**) front passenger’s feet resting spot, (**c**) rear left passenger’s feet resting spot, (**d**) rear right passenger’s feet resting spot.

**Table 1 sensors-23-01930-t001:** The parameters of measurement system.

Device:	Sensors		Data Acquisition Unit	
Name:	Sensor ADXL 204	Sensor ADXL 321	μDAQ 30	
Type:	accelerometer	accelerometer	A/D Channels	16/32 se
Measurement range:	±1.7 g	±18 g	A/D Resolution	16-bit
Nonlinearity:	0.2% FS	0.2% FS	A/D Sampling Rate	250 kHz
Sensitivity:	620 mV/g	57 mV/g	A/D Voltage Range	±10 V
Frequency band:	2.5 kHz	2.5 kHz	D/A Resolution	16-bit

**Table 2 sensors-23-01930-t002:** Selected frequency bands.

Frequency Band No [-]	Passband Start [Hz]	Passband Stop [Hz]
1	2	4.5
2	5	7
3	12	17
4	21	22
5	33	38
6	62	66
7	85	88
8	105	110
9	150	155
10	190	196

## Data Availability

The data presented in this study are available on request from the corresponding author. The data are not publicly available due to company policy.

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
