# Peer review of "Impact and Assessment of Suspension Stiffness on Vibration Propagation into Vehicle"

_sensors, 2023, doi:10.3390/s23041930_

Round 1

Reviewer 1 Report

I consider a completely objective and exhaustive review of the existing research on the subject of car suspension vibrations and their influence on the vibrations of the passenger area, and thus the impact on passenger comfort, to be a difficult task due to the large number of studies carried out all over the world on this subject. This makes the author's choice of literature all the more relevant and up-to-date. Perhaps it would be possible to select equally relevant other 40 literature items, but let us leave that to other authors.

The presented measurement results and their analysis are shown clearly and comprehensibly to the reader. However, I think that it could have been even better if the readability of some of the illustrations had been slightly improved.

Thus, figure No. 7 - I propose to show all channels in the same frequency range (0-250Hz) and in figures of the same size as the first 3.

Figure #8 - I propose to show all 5 diagrams in the same size, one below the other.

Figure #3 - I don't think it is by the author of the article, I suggest adding the source in the citations.

And a general remark, maybe showing all spectra on a logarithmic scale on the vertical axis would improve the visual recognition of the different signals? But here I leave the choice to the author.

I find the other illustrations and their description clear.

Author Response

Responses to the Reviewer

Thank you very much for your very insightful reviews and very valuable suggestions that allowed us to improve my manuscript and will probably influence my future research.

I have prepared a list of changes for each point that is being raised. It describes in details how I have incorporated the comments of the reviewers. In addition, I have submitted a version that highlights the changes I have made (marked version).

The contents of the review reports and responses are attached in pdf file.

Reviewer 2 Report

1. In abstract, the purpose of this paper is to study impact of spring stiffness parameters on the propagation of vibrations, most content of the abstract is the significance of this study. The results  of this work should be written clearly, and part of the innovation points and engineering significance can be put in introduction section.

2. A reread of the whole paper is necessary, trying to avoid very long and unclear periods. Such as “Studies of the impact exerted... in abstract.

3. Ambiguous statement (in introduction section): “In the case of research in the field of vibrations...motor vehicle. Therefore. It is difficult to follow the logic of authors writing and make sense of some of the sentences.

4. Please correct the punctuation in “ unsprung and sprung masses. [16,17]” and “ people in the vehicle. [18]”.

5. Figure 2 and 3 in your paper are a bit blurry. Please consider replacing them with clearer ones

6. In result and discussion, same problem, the results (figure 8 and 9) are too blurry to differentiate the legend.

7. The last three paragraphs of the conclusion section are about the significance of this study, and it would be better if they were more concise.

Author Response

(The authors gave the same response as above.)

Reviewer 3 Report

This work investigated the impact of spring stiffness parameters on the propagation of vibrations in the vehicle structure using simple amplitude and frequency measures. This research focused on the testing of actual suspension components with real damage under controlled conditions, the identification of the vibration propagation path, the quantitative comparison of vibration affecting humans in the vehicle, and the frequency decomposition of vibration for selected bands. It is a very useful and practical research, and interesting to readers. I suggest this manuscript can be accepted after minor revision. My specific comments are as follows,

-What is the advantages of the proposed method for reducing the propagation of vibrations in the vehicle? Please give some quantitative analysis by comparison with other methods.

-What is the disadvantages of using the developed method of selective multi-criteria analysis  in this research?

-The introduction does not analyze enough the existing literature on the specific topic: there is no clear claim about what is original compared to what has been done by others on the same matter.

-Results are not informative. Comparisons with others’ works are not enough.

-The reference part can be improved a lot by adding more researches on sensing of transportation structures, for example, Optical Fiber Technology, https://doi.org/10.1016/j.yofte.2022.103129.

-The previous work conducted by the authors is short of explanation, especially the experiments. Please add more researches done by the authors themselves on this topic.

Author Response

(The authors gave the same response as above.)

Reviewer 4 Report

Author in this paper presents results of research on the impact of spring stiffness parameters on the propagation of vibrations in the vehicle structure using simple amplitude and frequency measures. Author compares results of two suspension springs: a new and worn-out. Obtained measurement results are very clear and very detail presented.

Text on Figure 2 is very difficult to read! Please correct text in Figure 2!

In Figure 3 please explain assembly elements with the implementation of leaders and the name of the elements. This will help that the image become more understandable!

Please describe measuring equipment used in this experiment (technical characteristics, schemes, please use figures of this equipment in description)!

In chapter 3 (Results and discussion) author wrote: "This is due to the large number of articles in which the use of advanced and often very time-consuming mathematical algorithms and signal processing methods is presented". It would be good to reference some of those articles!

 PAPER NEEDS REVISION!

Author Response

(The authors gave the same response as above.)

Reviewer 5 Report

This article proposes using accelerometers to measure and record vehicle suspension and body vibration. Still, the technical method should be a standard method, and the experimental design, conditions, parameters, etc., need to be strengthened and supplemented. Please correct and resubmit.

The safety mentioned in the preface of the argument seems to be only about comfort. Please reconsider the research motivation.

If comfort is emphasized, a human/dummy experiment should be added, and a sensor should be installed on it to record its vibration transmission.

The interior will absorb most of the vibration transmitted into the car body by the suspension. Please consider this point. How does this experiment compare to the scene with interiors?

In addition, the vehicle is designed to continue driving even if the suspension springs fail. Please also consider research motivation.

The transmission of vibration is related to its medium. If the medium (such as structure {engine, interior}, material, joint method, etc.) is different, the transmission of its analysis may have no reference value.

If it is measuring equipment that can be provided, its installation, layout, and recording ability are the primary considerations.

The layout in Figure 2 does not conform to the sensors of channel 2~8 described in the text, and the labels are unclear. Please remake and clearly label the physical diagram of each sensor and its position in the entire frame.

If the vehicle does not even have an interior, how can it be estimated that the vibration is transmitted to the soles of the passengers' feet?

Figure 3. Considering only the spring as a variable of the entire suspension system is unreasonable. The premise is that all suspensions except the spring must be a predictable linear system. But in fact, the damper is not a linear system.

The description (caption) of the picture must still follow the specification of the text. (missing period).

The experiment in Fig. 4 only includes two springs with different properties, and the samples are too few. In addition, modern civilian vehicles have at least four independent suspensions, and please list the spring strain data used by each of the four suspensions.

The experimental method and structure are too simple to understand the overall experimental process. Please include more detailed experimental steps and photos of the overall experimental appearance, including the car body, moving plate, etc.

Figure 5 channel2, if this is the measurement of the vehicle chassis, please add the measurement of the actuating plate and the original output spectrum of the power source of the actuating plate.

The data is weak, and the correspondence between the external force input and the channel is unclear. For example, whether channel 2 is a passive vibration or an input vibration, and the information of the input vibration source is incomplete.

Figure 5 only a single data does not conform to the spirit of the experiment. The data is poorly presented and cannot make sense visually.

Please include the displacement data, the feeling caused by the vibration is correlated with the displacement.

page 8, the last phase, it seems that it is not accurate enough to discuss the vibration transmitted from the vehicle to the soles of the passengers.

After all, passengers will ride in the condition of interior decoration, seats, and shoes, and the vibration mainly transmitted to passengers is from the hips to the back and then the feet and hands. Please modify the appropriate argument basis.

Suspension is a matter of comfort, not safety.

Author Response

(The authors gave the same response as above.)

Round 2

Reviewer 4 Report

Dear Authors,

after you have expanded your research with additional descriptions and clarifications and in this way have additionally contributed to increasing the quality of already exceptionally high-quality research; I consider your work original and acceptable!

Best regards!

Author Response

Thank you very much for your very insightful reviews and very valuable suggestions that allowed me to improve my manuscript and will probably influence my future research.

The review report allowed me to critically look at the manuscript, carefully analyze it, and make significant corrections. Due to this, the article has been greatly improved.

Thank you very much for the review and positive opinion.

Reviewer 5 Report

The authors have answered all of my questions, and the paper has significantly improved. Therefore, it can be accepted for publication.

I have some suggestions, and they will not affect the acceptance of this article. The presentation of experimental data in this paper should have a more straightforward visual way, such as in Figure 6 and Figure 7, as well as other graphs. There are also, for example, Fig. 11, Fig. 12, and so on. Table 1 Please recheck whether the item correspondence is appropriate.

Author Response

Thank you very much for your very insightful reviews and very valuable suggestions that allowed me to improve my manuscript and will probably influence my future research.

The review report allowed me to critically look at the manuscript, carefully analyze it, and make significant corrections. Due to this, the article has been greatly improved.

Thank you very much for the review and positive opinion.

I fully agree that presentation of experimental data should have straightforward visual way. The current content and presentation of experimental data result from the comments of all reviewers (five reviewers evaluated the manuscript). It was fully accepted by the other reviewers, so subsequent changes may affect those that I made according to the suggestions of other reviewers.

I did rechecked Table 1 and I can confirm that the items correspondence is appropriate.

For confirmation, I attach the reviewer reports of all Reviewers in the previous iteration (Round 1) and my responses and corrections. In round 2, all reviewers accepted the article for publication in its current form.
